# HPLC–(Q)-TOF-MS-Based Study of Plasma Metabolic Profile Differences Associated with Age in Pediatric Population Using an Animal Model

**DOI:** 10.3390/metabo12080739

**Published:** 2022-08-11

**Authors:** Oihane E. Albóniga, Oskar González-Mendia, María E. Blanco, Rosa M. Alonso

**Affiliations:** 1Department of Analytical Chemistry, Faculty of Science and Technology, University of the Basque Country (UPV/EHU), Barrio Sarriena s/n, 48940 Leioa, Spain; 2Metabolomics Platform, CIC bioGUNE, CIBERehd, Bizkaia Technology Park, 48160 Bilbao, Spain; 3Painting Department, Faculty of Fine Arts, University of the Basque Country (UPV/EHU), Barrio Sarriena s/n, 48940 Leioa, Spain

**Keywords:** metabolic profiles, animal model, mass spectrometry, glycerophospholipids, pediatric population

## Abstract

A deep knowledge about the biological development of children is essential for appropriate drug administration and dosage in pediatrics. In this sense, the best approximation to study organ maturation is the analysis of tissue samples, but it requires invasive methods. For this reason, surrogate matrices should be explored. Among them, plasma emerges as a potential alternative since it represents a snapshot of global organ metabolism. In this work, plasma metabolic profiles from piglets of different ages (newborns, infants, and children) obtained by HPLC–(Q)-TOF-MS at positive and negative ionization modes were studied. Improved clustering within groups was achieved using multiblock principal component analysis compared to classical principal component analysis. Furthermore, the separation observed among groups was better resolved by using partial least squares-discriminant analysis, which was validated by bootstrapping and permutation testing. Thanks to univariate analysis, 13 metabolites in positive and 21 in negative ionization modes were found to be significant to discriminate the three groups of piglets. From these features, an acylcarnitine and eight glycerophospholipids were annotated and identified as metabolites of interest. The findings indicate that there is a relevant change with age in lipid metabolism in which lysophosphatidylcholines and lysophoshatidylethanolamines play an important role.

## 1. Introduction

Drug absorption, distribution, metabolism, and excretion (ADME) are substantially different in children as compared with adults due to time-related development and organ system maturation processes [1]. Children’s bodies grow and change rapidly during early life, especially during the first 2 years [2]. In spite of the increase in available information related to drug absorption and disposition in the pediatric population, the impact of specific age-related effects on pharmacokinetics (PK) and pharmacodynamics (PD) remains poorly understood [2]. The main reasons for this lack of knowledge are the scarce number of clinical trials, the heterogeneity of the pediatric population, and ethical concerns [3]. As a consequence, many drug therapies used in children are unlicensed or prescribed off-label [4,5], calculating the dose to administer by an estimation of that used in adults, corrected by the weight or volume of the patient [6,7]. However, the multiple physiological and anatomical differences generate disparities in ADME processes between children and adults [8]. To solve this issue, a better understanding of human developmental biology is needed. In this context, a deeper insight into organ maturation would help to achieve a more effective and safer drug therapy in accordance with children’s biological age.

There is an increasing interest in deeply understanding the biological mechanisms both at the molecular level and in the organism as a whole. For this purpose, metabolomics is a powerful bioanalytical strategy to explore the current state of the metabolites involved in a biological process, such as the different phases that occur during organ development at early life stages. Metabolomics is based on the study of the complete set of metabolites in a biological sample. In this way, it is a promising tool to improve the knowledge on age-dependent organ maturation by comparing metabolic profiles from children of different ages. Although metabolomics has been applied in several areas, such as disease prognosis, treatment efficiency, and nutrition [9], the number of studies related with the growth or development of children is scarce [10]. Considering the difficulties in obtaining samples from pediatric population, the use of animal models has been a recurrent alternative. In this aspect, minipigs or piglets have demonstrated to be a suitable animal model of children and are widely used in pediatric studies due to their similarity in size, physiology, organ development, and so on [11,12]. To the best of our knowledge, only two works have explored the differences in the metabolic profiles associated with age: one of them in urine samples collected from neonates during the first 4 months of life [13] and the other one in piglets’ plasma samples [12].

Metabolic profiles can be obtained from different biological matrices, being the tissue samples the most appropriate for organ maturation studies. Nevertheless, the difficulties in taking biopsy materials lead to the use of surrogate samples, such as blood or derived biofluids (plasma and serum) and urine [14]. The collection of these biofluids is easier, less invasive, and more affordable, especially for large-scale studies. Furthermore, they can be analyzed after simple sample treatments, such as protein precipitation or sample dilution [15,16,17]. In this sense, blood and its derived biofluids are matrices of special interest because many compounds are released into them by different organs. These compounds travel to and from specific tissues, or they are secreted from cells, and normally they are recycled or excreted to the blood after their metabolism. This issue is decisive as they represent a potential source of information about the physiological state of an individual and provide a picture of the global organ or tissue metabolism [18].

Taking into account the huge amount of metabolites in biological matrices, discovering biomarkers and identifying the significant features is a big challenge [19,20]. Even though multivariate analysis, such as principal component analysis (PCA), has demonstrated to be a useful tool in metabolomics for clustering and data quality assessment, new chemometric approaches have been proposed lately. In this way, multiblock methods and correlation analysis are useful for integrating data generated by different experiments or conditions, such as positive and negative ionization modes in mass spectrometry.

The aim of this work was the application of metabolomics to study and compare the metabolic profiles of plasma samples obtained from three groups of piglets of different ages (newborns, infants, and children) in order to find putative biomarkers that can be related to organ maturation state. For this purpose, high-performance liquid chromatography coupled to a hybrid quadrupole-time-of-flight mass spectrometer (HPLC–(Q)-TOF-MS) system was used at positive and negative ionization modes. Multivariate and univariate statistical analyses were used for finding significant features in plasma that may reflect the organ maturation state. Multiblock principal component analysis (MB-PCA) and Pearson correlation studies were used as complementary tools to assess sample clustering and deal with metabolite annotation challenge, respectively.

## 2. Materials and Methods

### 2.1. Reagent and Solutions

The LC–MS grade formic acid and acetonitrile (ACN) used for the mobile phases’ preparation were purchased from Fisher Scientific (Pittsburgh, PA, USA) and Scharlab, S.L. (Barcelona, Spain), respectively. The methanol (MeOH) used in the sample and reagent solution preparation was obtained from Scharlab, S.L. (Barcelona, Spain). In addition, ultra-high-purity water, obtained from tap water pre-treated by Elix reverse osmosis and a Milli-Q system from Millipore (Bedford, MA, USA), was used for mobile phase, reagent solutions, and sample preparation.

Standard reagents used to assess the proper LC–MS system operation were from different manufacturers: paracetamol, cholic acid, (±) verapamil hydrochloride, simvastatin, reserpine, and leucine enkephalin acetate salt hydrate were provided by Sigma-Aldrich (Steinheim, Germany); caffeine and salicylic acid were supplied by Alfa Aesar (Karlsruhe, Germany) and Fluka Analytical (Bucharest, Romania), respectively. Finally, sodium fluvastatin was kindly supplied by Novartis (Basel, Switzerland). A system suitability test solution (SST) was prepared with the nine compounds at a final concentration of 100 ng/mL in MeOH:H_2_O 2:1 (*v*/*v*); this solvent composition was chosen as this is similar to the final solution (supernatant) of plasma samples.

### 2.2. Study Design and Sample Collection

Plasma samples were collected by the team of the Experimental Neonatal Physiology Unit of the BioCruces Health Research Institute (Cruces University Hospital, Basque Country, Spain), following the European and Spanish regulations for the protection of experimental animals (86/609/EFC and RD 1201/2005), and the procedure was approved by the Ethical Committee for Animal Welfare. Samples were obtained from mechanically ventilated newborn piglets, or group A (<5 days, *n* = 12); infant piglets, or group B (2 weeks, *n* = 12); and child piglets, or group C (4 weeks, *n* = 12) of Topig F-1 Large White × Landrave breed. Each group contained the same number of females and males. Whole blood samples were collected in K_2_-EDTA tubes, and they were immediately centrifuged at 950× *g* for 10 min at room temperature in order to obtain plasma. The supernatant was transferred to a cryovial and stored at −80 °C until analysis.

### 2.3. Plasma Sample Treatment and QC Sample Preparation

Frozen plasma samples were thawed at room temperature, and protein precipitation was carried out with 50 µL of plasma and 100 µL of cold MeOH. After vortex mixing for 2 min in a Signature Digital Vortex Mixer 945303 (VWR, Radnor, PA, USA), samples were centrifuged at 16,110× *g* for 15 min at 10 °C in a 5415R Eppendorf centrifuge (Hamburg, Germany). The clean upper layer was transferred to a chromatographic vial to be injected into the HPLC–(Q)-TOF-MS system.

A quality control sample (QC) was prepared by taking 5 µL of each plasma sample. After vortex mixing, 50 µL was taken and treated as previously described. The QC sample was injected at the beginning of the run to equilibrate the system and then every six randomized samples. These QCs were used to assess the reproducibility and stability of the system and when necessary for signal correction within the analytical sequence.

### 2.4. HPLC–(Q)-TOF-MS Analysis

Metabolomics analysis of plasma supernatants was performed using a 1200 series HPLC system coupled to a 6530 series hybrid quadrupole time-of-flight mass spectrometer (Q-TOF-MS) from Agilent Technologies (Santa Clara, CA, USA), equipped with an Agilent Jet Stream electrospray (ESI) source. Samples were randomized along the run to reduce any time-related effects. Chromatographic separation was carried out injecting 5 µL of plasma supernatant on a Zorbax SB-C18 reversed phase chromatography column (2.1 × 100 mm, 3.5 µm) equipped with a C8 guard column (2.1 × 12 mm, 5 µm), both from Agilent Technologies, at 35 °C and a flow rate of 0.4 mL/min. A binary solvent system consisting of 0.1% formic acid and 5% ACN in water (phase A) and 0.1% formic acid in ACN (phase B) was used for the elution. The gradient started from 0 to 100% B over 10 min, remained at 100% B for 2.5 min, returned to starting conditions in 1.5 min, and re-equilibrated for 5 min. The mass spectra data were acquired firstly in positive and then in negative ionization modes with capillary voltages of +3800 V and −2500 V, respectively. The other ESI source parameters were set as follows: dry gas (nitrogen) temperature, 325 °C; dry gas flow, 10 L/min; nebulizer gas (nitrogen) pressure, 30 psig; sheath gas temperature, 350 °C; sheath gas flow, 11 L/min; skimmer, 65 V; fragmentor, 125 V, and octopole RF peak, 750 V.

The MS detector operated in a low mass range (<1700 *m*/*z*) and a 2 GHz extended dynamic range, and centroid acquisition mode was used for data collection and storage. A reference solution was directly infused into the source to ensure mass accuracy, reproducibility, and continuous internal calibration during the analysis. Two reference masses for each ionization mode, at *m*/*z* 121.0509 (purine, [C_5_H_4_N_4_+H]^+^) and *m*/*z* 922.0098 (HP-921, [C_18_H_18_O_6_N_3_P_3_F_24_+H]^+^) for the positive mode and *m*/*z* 112.9855 (TFANH_4_, [C_2_H_4_O_2_NF_3_-NH_4_]^−^) and *m*/*z* 966.0007 (HP-921COOH, [C_18_H_18_O_6_N_3_P_3_F_24_-COOH]^−^) for the negative mode, were used during the HPLC–(Q)-TOF-MS run. Firstly, the analysis was carried out in the MS scan mode, where isolation or fragmentation was not applied, and all the ions were conducted through the quadrupole. For this acquisition mode, the mass data was collected at a scan rate of 2 scans/s between *m*/*z* 50 and 1200. Then, MS/MS analysis was carried out for the compounds of interest at a scan rate of 5 scans/s using *m*/*z* dependent collision energies: 25 V for *m*/*z* lower than 300, 30 V for *m*/*z* between 300 and 850, and 35 V for *m*/*z* greater than 850. Additionally, the SST and a blank sample (MeOH:H_2_O, 2:1, *v*:*v*) were injected in the HPLC–(Q)-TOF-MS at the beginning, in the middle, and at the end of each sequence in order to control the analytical performance (HPLC system and MS instrument) and detect and/or remove artifacts, interferences, or pollutants in the solvents.

The data were acquired using the Agilent MassHunter Workstation version B.05.01, and the raw data were processed with the MassHunter Qualitative version B.07.00, both from Agilent Technologies.

### 2.5. Data Preprocessing

Metabolomic data were treated to reduce the complexity and to obtain a two-dimensional table (matrix) with a list of features (pairs of mass-to-charge ratio (*m*/*z*)–retention time (RT)) with their intensities. Firstly, LC–MS raw data were checked using the MassHunter Qualitative software to determine the chromatographic performance quality and the system mass accuracy as well as QC injection reproducibility and pressure stability. Then, raw data were converted into the open format mzXML using msConverter (proteoWizard) [21] from 0 to 13.5 min so that features coming from the cleaning step of the elution gradient were removed. The detection of features was carried out in R 3.4.3 (https://www.r-project.org/; accessed on 11 November 2019) using the XCMS 1.52.0 package (Metlin, La Jolla, CA, USA) [22,23]. The algorithm parameters were optimized with the freely available Isotopologue Parameters Optimization (IPO) [24] package following the criteria reported by Alboniga et al. [25], and they are displayed in Table 1. CentWave was the algorithm employed for peak detection, the obiwarp algorithm was used for retention time correction, and the density algorithm was used for grouping. Afterward, and in order to avoid zero values in the resulting matrix that could bias the statistical analysis, a filling step was carried out to force the integration of peaks where no signal was detected. Finally, CAMERA 1.32.0 package (Bioconductor Open Source Software for Bioinformatics) [26] was used for isotopologues and adducts detection. The resultant matrix for each ionization mode was treated and filtered before statistical analysis as follows: the isotopes identified by CAMERA ([M + 1], [M + 2], and [M + 3]), the features before the injection peak (less than 1 min), and the features with percentage of relative standard deviation (%RSD) in the QCs greater than 20% were removed [27].

### 2.6. Multivariate Analysis

Plasma data matrices obtained for positive (Plasma ESI+) and negative (Plasma ESI−) ionization modes in the previous step were further processed using MATLAB software (The MathWorks, Naticks, MA, USA) and the toolbox freely available online at https://github.com/Biospec/cluster-toolbox-v2.0; accessed on 11 November 2019. Intensity drop was corrected with the QC correction function included in the toolbox, and then, autoscaling or logarithm (log_10_) scaling was applied to avoid experimental variations that may cause differences in orders of magnitude among different metabolites [28].

Multivariate analysis was performed on the scaled matrices to explore the relationship among samples. For this purpose, PCA was employed to reduce the dimensionality of the data [29,30] and the PCA score plot was analyzed to determine the group clustering, the data quality, and the presence of outliers [31].

Furthermore, partial least squares-discriminant analysis (PLS-DA), a particular case of the PLS algorithm, was used to model the relationship between the measured features and the target class label (newborn, infant, and child piglets) when clear clustering was not achieved by PCA modeling. PLS-DA has the main advantage of handling collinear and noisy data, which are common outputs from metabolomics experiments [30]. Moreover, the PLS-DA method also helps to solve classification problems by maximizing the separation among the classes through extracting the latent variables [32,33]. However, the overoptimistic nature of the PLS-DA classification method, also known as overfitting, requires a suitable validation method to obtain an appropriate and reliable classification model [34,35]. For this purpose, bootstrapping with replacement (1000 interactions) was used to generate the multiple training and test data sets (splitting the data) [34] in combination with permutation testing (1000 permutations), where sample labels are randomly permuted and a new classification model is calculated [36]. The double-check validation method generates an average confusion matrix and a correct classification rate (CCR) graph to evaluate the distribution of a random classification and assess the statistical significance of the model by considering all the possible permutations (*p-*value) [36].

Finally, multiblock modeling, a useful method designed to find the underlying relationship between data matrices of possible related data sets, was used to improve group clustering compared to classical PCA and to study the correlations between the positive and negative data sets. For this aim, the plasma data obtained in positive and negative ionization modes were joined into a single multiblock data structure to perform MB-PCA modeling [37,38]. The algorithm employed for this purpose was the CPCA-W proposed by Westerhuis et al. [39]. The filtered matrices obtained in the preprocessing step were further treated to build the MB-PCA model. First, QC correction was done and QC samples were removed. Afterward, autoscaling was applied and the square root of the number of variables in each block (1/√*n*, *n* being the number of variables in the block) was used to avoid dimensionality differences between matrices [37]. Then, the multiblock matrix was used to perform the MB-PCA analysis in order to evaluate the clustering within groups when different ionization modes were joined together.

### 2.7. Univariate Analysis

In order to obtain the significant features, univariate analysis was carried out to find features that discriminate the three groups of piglets under study (newborns (A), infants (B), and children (C)). For this purpose, parametric tests (one-way ANOVA with a false discovery rate (FDR)) were applied to find significant differences among the studied groups. Since this test does not establish where the difference lies, a multiple comparison test (post-hoc Tukey HSD (honestly significant differences)) was applied in order to select only those features that are different among the three groups of piglets (A ≠ B ≠ C). Afterward, as ANOVA works under normality assumption, Lilliefors test was used to study the normal distribution. In all those features that did not show a normal distribution, the Kruskal–Wallis non-parametric test was applied to study differences among the studied groups. Therefore, a feature was considered significant when the *p*-value was lower than 0.001 and fulfilled the criteria for the post-hoc Tukey HSD test (A ≠ B ≠ C).

### 2.8. MS/MS-Based Metabolites Annotation

All the significant features obtained in the univariate analysis were further studied to determine the peak quality. For this purpose, extracted ion chromatograms (EICs) were obtained with the MassHunter Qualitative software incorporated in the HPLC–(Q)-TOF-MS system and peak shapes were studied. Only those peaks clearly differentiated from the noise signal were considered for further experimental fragmentation analysis.

Then, Pearson pair-wise correlation studies were performed in the scaled matrix in order to facilitate metabolite annotation and to obtain information in both ionization modes. Thereby, a significant feature in one ionization mode, which also distinguishes the three groups, had to be detected in the other ionization mode with a correlation coefficient greater than 0.8 [37]. Finally, two features coming from the same molecule have to elute at same RT. Thus, among those previously selected features, only those with a comparable retention time in the ESI+ and ESI− data runs were chosen. The time window used to consider that two features elute at the same retention time was two times the maximum variability in the SST compounds.

Finally, the features selected in the previous steps were analyzed by tandem MS in order to obtain their fragmentation pattern. The fragmentation experiments were performed under the same chromatographic conditions as MS scan analysis and using the MS/MS conditions described in Section 2.4. Metabolite annotation was based on the comparison of the experimental mass obtained in the MS scan analysis with the theoretical accurate mass in online databases. Furthermore, the isotopic pattern was studied and compared with the available databases, such as METLIN [40], Human Metabolome Database (HMDB) [41], mzCloud [42], MyCompoundID [43], and LIPID MAPS [44]. A mass tolerance below 5 ppm was accepted. Then, the molecular feature extraction (MFE) algorithm provided by the MassHunter Qualitative Analysis software was also used to obtain a possible formula in order to have a starting point and improve the metabolite annotation. MFE considers the accurate mass, isotopic patterns, relative abundances, and *m*/*z* distances to obtain a possible formula with an average ID score (%) [45]. Afterward, the experimental MS/MS spectra were compared with the MS/MS spectra information available in the databases to verify the coincidence by comparing the fragmentation. Finally, some metabolites that were not in the databases were annotated combining fragmentation pattern study with literature searching.

## 3. Results

### 3.1. Multivariate Analysis

After preprocessing the converted mzXML raw data with XCMS and filtering the matrices to remove the injection peak features, the isotopes, and the non-repeatable features (%RSD > 20 in the QCs), 2207 features in Plasma ESI+ and 1855 features in Plasma ESI− were obtained. Autoscaling was used for Plasma ESI+ data set and log_10_ for Plasma ESI− before PCA modeling in order to study the quality of the analysis and determine group clustering. Afterward, the QC correction function incorporated in the MATLAB toolbox was applied to both data sets and PCA was performed again (see Appendix A).

Once adequate results were achieved, the QC group was removed from the PCA in order to assess the group clustering. As shown in Figure 1, the three groups (newborns (A), neonates (B), and infants (C)) reveal a clear tendency in the PC1 even though they are not totally separated and some groups overlap. In Plasma ESI+, group C is separated from B and A, whereas in Plasma ESI−, group A is the one clearly separated from B and C.

Due to the overlapping observed between groups A and B in Plasma ESI+ and between groups B and C in Plasma ESI−, supervised analysis by PLS-DA was performed. In Figure 2, the PLS-DA score plot models are shown, where a clear separation among groups was found.

Owing to the overfitting nature of the PLS-DA method, a proper validation was performed to determine if the classification was fortuitous. For this purpose, bootstrapping with replacement with 1000 interactions was applied for splitting the data. The bootstrapping method was combined with permutation testing (*n* = 1000), which assessed if the real classification was better than any other possible classification. The resultant average confusion matrices and correct classification rate (CCR) graphs are shown in Appendix A for Plasma ESI+ and Appendix A for Plasma ESI− (see Appendix A). More than 90% of the samples were correctly allocated except for infants (B) in Plasma ESI+ and children (C) in Plasma ESI−. The misassignments observed for these groups are consistent with the distribution shown in the PCA score plots (Figure 1). Finally, the CCR results show that the real classification (observed distribution) is significantly better than any other random classification (null distribution) with *p*-value less than 1.0 × 10^−3^. This means that the classification models were validated and the classification is not fortuitous.

Finally, MB-PCA modeling was performed using the filtered matrices of 2207 features for Plasma ESI+ and 1855 for Plasma ESI−. After autoscaling the matrices, which is required for MB-PCA modeling, block-weighting was applied. Then, the data were arranged into two blocks: the first block contained all the samples in positive ionization mode (Plasma ESI+ or Block 1) and the second block in negative ionization mode (Plasma ESI− or Block 2). The resultant MB-PCA super score plot, which represents the common trend across Plasma ESI+ and Plasma ESI− matrices [37], is shown in Figure 3, and the block score are shown in Figure 4 (upper score plots). In all cases, clear separation between group A and the other groups (B and C) was obtained but groups B and C were still partially overlapping.

In order to assess the clustering improvements when data were fused, the block score PCAs (upper score plots in Figure 4) were compared with the classical PCA (lower score plots in Figure 4). As can be seen, the Plasma ESI+ data set has similar clustering in both cases, with the tendency appearing mainly in the PC1. In the case of Plasma ESI−, a new classical PCA was performed after applying autoscaling (initially log_10_ was used) and QC correction in order to properly compare the clustering among groups with that observed in the block score PCA (right score plots in Figure 4). As can be seen, in both cases, group A was clearly separated in the PC1 from groups B and C. However, infants (B) and children (C) clustered together when classical PCA was performed. Even though total clustering was not achieved in Plasma ESI−, an improvement was observed when data were fused.

### 3.2. Univariate Analysis

The univariate analysis pipeline described previously was followed in order to find significant features that can explain the differences between different groups of samples. It is important to highlight that univariate analysis was applied in the filtered matrix used for PCA building as PLS-DA was only employed for classification modeling. In Table 2, the number of features obtained in each step of the pipeline is shown. Finally, the number of significant features obtained was 27 and 74 for Plasma ESI+ and Plasma ESI−, respectively.

### 3.3. MS/MS-Based Metabolite Annotation

The *m*/*z* values of the significant features obtained after the univariate analysis in Plasma ESI+ and Plasma ESI− data sets (Table 2) were extracted from the chromatogram using the MassHunter Qualitative software in order to determine if they were real peaks or artifacts. Among the total significant features, 13 out of 27 in Plasma ESI+ and 21 out of 74 in Plasma ESI− were considered to be real peaks. Then, Pearson correlation analysis was performed and the correlation coefficients were calculated. Those pairs of features formed by at least one feature that distinguishes the three groups of piglets (A ≠ B ≠ C) in a significant way (*p*-value < 0.001) and with a correlation coefficient greater than 0.8 were considered. In some cases, the pairs of features ionized in one mode could be paired with several features in the other ionization mode. This means that they may belong to the same metabolite, as adducts or isotopes not identified by CAMERA or in-source fragments of that metabolite.

Finally, MS/MS experiments were carried out to obtain the fragmentation pattern of those features selected from the univariate analysis (see Table 3) and the correlation studies (see Appendix A). In Table 3, the significant features obtained from the univariate analysis that were considered real peaks are displayed with their *m*/*z* value, RT, trend within groups, and putative annotation based on MS/MS studies. Finally, those pairs of metabolites after Pearson correlation studies are included in Appendix A.

In order to annotate the metabolites, the experimental accurate mass and the MS/MS spectra were compared with those available in METLIN, HMDB, mzCloud, MyCompound ID, and LIPID MAPS. The lack of many MS/MS spectra in databases with an acceptance mass error < 5 ppm for our selected compounds led to deeper investigations to annotate metabolites of interest. In this sense, the MS/MS fragmentation pattern was studied through the significant features to obtain information about class types (see Table 3).

The study of the MS/MS patterns in Plasma ESI+ allowed the discovery of a feature with *m*/*z* 184.0718, which was common among several metabolites. The *m*/*z* 184.0718 is known to be the protonated phosphocholine (C_5_H_15_NO_4_P), which is a characteristic fragment of compounds with a phosphocholine head group, such as phospholipids (PL) and glycerophospholipids (GPLs). The fragmentation pattern of the protonated phosphocholine found in METLIN was used and compared with the experimental MS/MS of the metabolites of interest, in order to classify or annotate them. For instance, the fragmentation pattern obtained experimentally (see Figure 5) for the significant metabolite with *m*/*z* 544.3400 and RT 10.54 min confirms that it belongs to a lipid class with a phosphocholine head group. In order to annotate the lipid class of this feature, LIPID MAPS [44] database and literature [46] were used and it was annotated as lysophosphatidylcholine (LPC) C20:4 (LPC(20:4)), which means that it has a monounsaturated fatty acid with 20 carbons as well as the phosphocholine head group. Following the same procedure, another two metabolites were annotated as LPCs: LPC (20:2) with *m*/*z* 548.3703 and RT 11.52 min (see Appendix A) and LPC class with *m*/*z* 300.6346 and RT 10.55 min (see Appendix A).

It is important to point out that the metabolite with *m*/*z* 530.3244 at RT 11.04 min (Figure 6) could initially be assigned to two lipid classes, lysophoshatidylethanolamine (LPE) and LPC. This metabolite was putatively annotated as LPE (22:4) [M+H]^+^ because the MS/MS contained the *m*/*z* 389.3089, which fits with the loss of the phosphoethanolamine (*m*/*z* 141.0165) [46]. The other putative annotation given (LPC (17:1) [M+Na]^+^) has two fragments of the phosphocholine head group (*m*/*z* 104.1046 and *m*/*z* 66.0950) as well as the characteristic *m*/*z* 184.0681. The lack of the sodiated phosphocholine fragment ion (*m*/*z* 146.9817), characteristic of the fragmentation of sodium adducts of LPCs, made this annotation less reliable but still feasible. In order to elucidate this metabolite as LPE (22:4) or LPC (17:1), the correlated feature in negative ionization mode (*m*/*z* 528.3077 and RT 11.01 min), obtained by Pearson correlation analysis, was studied. The *m*/*z* 528.3077 of this feature was annotated as LPE (22:4) [M-H]^−^ by LIPID MAPS with less than 5 ppm of error. Thus, Pearson correlation analysis facilitated metabolite annotation and demonstrated to be a complementary useful tool.

Furthermore, among the significant features in positive ionization mode, an acylcarnitine class feature was found. The comparison of the MS/MS spectra from METLIN of acetyl-carnitine with the MS/MS obtained for the feature with *m*/*z* 356.2795 and RT 9.7 min showed the characteristic fragment of *m/**z* 85.0268 (see Appendix A). This means that this significant feature belongs to the acylcarnitine class. Although several studies deal with the characterization of acylcarnitines using MS, the annotated acylcarnitine in this work is not among those included in the literature [47,48,49,50].

The pipeline applied to the Plasma ESI+ significant features was also followed for the negative ionization mode in order to annotate the metabolites of interest. The MS/MS spectra for the annotated metabolites in negative ionization mode are shown in the Appendix A). All of them were annotated based on the fragmentation pattern described by Godzien et al. [46] and the mass coincidence with LIPID MAPS. The annotated metabolites are displayed in Table 3.

The metabolite of interest with *m*/*z* 586.3140 at RT 10.03 min (see Appendix A) was annotated as LPC (20:5) ([M+COOH]^−^) due to the presence of an ion (*m*/*z* 526.2992) resulting from the loss of 60.0222 (methyl formate), which is the main characteristic fragment for the recognition of this adduct [46]. Then, three other metabolites were annotated: one as an LPC class with *m*/*z* 615.3475 and RT 10.69 min (see Appendix A), one as an LPE (18:1) (see Appendix A) with *m*/*z* 478.2922 and RT 10.89 min, and one as an LPE (15:1) (see Appendix A) with *m*/*z* 436.2815 and RT 10.98 min. 

## 4. Discussion

Taking into account the annotated metabolites, further literature search was carried out to determine the biological role of this class of metabolites. The annotated metabolites belong mainly to different lipid species. Indeed, this complex family of biomolecules has been associated with several biological pathways, some of which are significantly modified with age.

There have been reported more than 600 distinct molecular species covering the six main mammalian lipid categories (sterol lipids, glycerophospholipids, glycerolipids, sphingolipids, fatty acids, and prenol lipids) [51]. Among them, GPLs enclose a high proportion of total lipids present in plasma and over 200 species were detected in the human plasma standard reference material (SRM) [51,52]. This family is of special interest in this study because most of the annotated metabolites (LPEs and LPCs) belong to lysophospholipids (LPLs), which are substrates or intracellular products of GPLs [53].

LPLs are small bioactive lipid molecules containing a single fatty acyl chain and a polar head group. The diversity on LPLs is based on the polar head group and the fatty acyl chain coupled to the glycerol backbone. At the same time, certain LPL subclasses, such as LPEs and LPCs, can have different fatty acyl chains linked. The intracellular production of LPLs from GPLs consists in the de-acylation by hydrolysis of the carboxyl ester bonded at the *sn*-1 or *sn*-2 position by the action of the phospholipase A_2_ to release the fatty chain. This is further re-acylated by lysophospholipid acyltransferase (LPLATs) to generate a new specie, which can be different from the original de-acylated one. This process is known as Lands’ cycle or the remodeling pathway and involves several enzymes. In the case of LPLs in plasma, they can be generated by the action of lecithin:cholesterol acyltransferase (LCAT), which is expressed and secreted primarily from the liver. LCAT, a phospholipase A_2_ enzyme, is related to lipoprotein metabolism and might be susceptible to present changes in the expression related to the organ maturation state in liver.

Thus, the significant difference observed in several LPC and LPE levels (see Appendix A) between newborn, infant, and child piglets could be explained by the immaturity or different states of activity in the enzymes involved in Lands’ cycle or in the expression of the LCAT. It is known that both GPLs and LPLs are involved in several biological functions, such as the source of energy, cell membrane components, and cellular signaling messengers [54]. Furthermore, it is reported that LPLs increase under pathological conditions so they are gaining special interest as diagnostic and/or pharmaceutical markers [55,56]. However, no studies correlating GPLs or LPLs with children were found. Consequently, further studies are needed to obtain more information on and clarify the relationship of the LPCs and LPEs with the organ maturation state in the pediatric population.

In addition to the LPL compounds, an acylcarnitine class metabolite was annotated. The acylcarnitine did not match with available databases or the literature. The acylcarnitine follows a down-regulated tendency among newborn, infant, and child groups. The tendency of this annotated acylcarnitine (*m*/*z* 356.2795), which belongs to the acylcarnitines class with more than four carbons in the linked fatty acid (short-chain fatty acid) [57], is in agreement with Cavedon et al. [58], where the medium- and long-chain acylcarnitines decreased with age. However, the down-regulated tendency is not in agreement with the work of Novak et al. [59], in which, it was observed that total acylcarnitine levels grow with age.

## 5. Conclusions

The comparison between plasma metabolic profiles obtained from newborn, infant, and child piglets showed a clear tendency within groups. However, no clear clustering was achieved. The supervised analysis and its validation finally demonstrated that there are differences between the three groups of piglets and that the classification is not fortuitous (*p*-value < 1.0 × 10^−3^). Furthermore, the multiblock principal component analysis (MB-PCA) has demonstrated to be a promising tool to obtain complementary information and to improve the clustering within groups compared to classical PCA.

Univariate analysis identified 13 and 27 metabolites for Plasma ESI+ and Plasma ESI− data sets, respectively. These metabolites were considered for further MS/MS studies, which finally allowed the annotation of nine metabolites as putative biomarkers. Among them, the main class of annotated metabolites in plasma belongs to lipid classes, mainly LPCs (LPC (20:4), (20:2), and (20:5)) and LPEs (LPE (18:1), (15:1), and (22:4)), which are part of the second-most-abundant lipid class in plasma (GPLs) and were down-regulated or up-regulated among groups (see Appendix A). Furthermore, other lipids were found within the annotated metabolites but no class could be assigned. The absence of studies related with LPC and LPE tendency in children makes the findings in this study highly relevant to improve the knowledge of age-related development in the pediatric population.

In summary, annotated metabolites are considered to be putative biomarkers of the physiological state and global organ metabolism among piglets of different ages (newborns, infants, and children). Among them, lipids are found to be of special interest, which suggests that a lipidomics approach could be relevant for further studies. Certainly, other complementary techniques for the analysis of more polar molecules or volatile compounds would provide a broader picture of and new information about the changes occurring in pediatric metabolism. Moreover, as plasma could be considered the main transport of metabolites released to and from tissues, it is a feasible matrix to investigate other tissues and their maturation state with age. Therefore, the findings presented in this work could be considered an important starting point for further investigations on organ maturation and development process in children, which could lead to improvement in the knowledge on organ maturation development, which at the same time would lead to better drug dosing in the pediatric practice.

## Figures and Tables

**Figure 1 metabolites-12-00739-f001:**
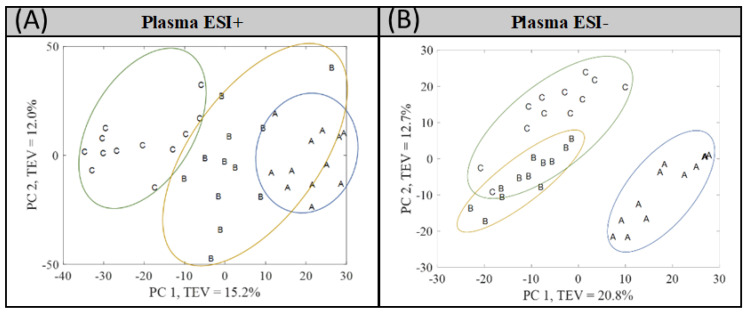
PCA score plots obtained after QC samples were removed for the matrices of (**A**) Plasma ESI+ and (**B**) Plasma ESI−. Piglet groups: newborns (A), infants (B), and children (C). TEV = total explained variance.

**Figure 2 metabolites-12-00739-f002:**
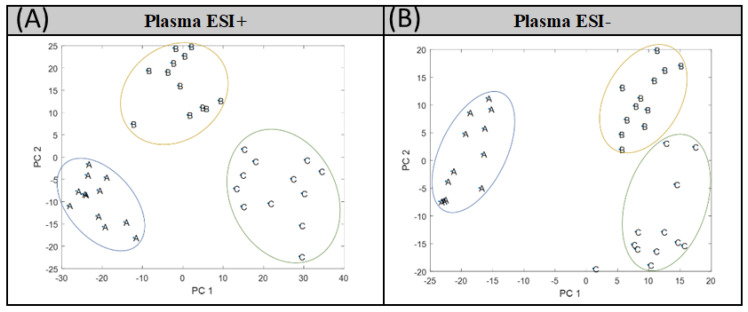
PLS-DA score plots obtained for (**A**) Plasma ESI+ and (**B**) Plasma ESI− matrices. Piglet groups: newborns (A), infants (B), and children (C).

**Figure 3 metabolites-12-00739-f003:**
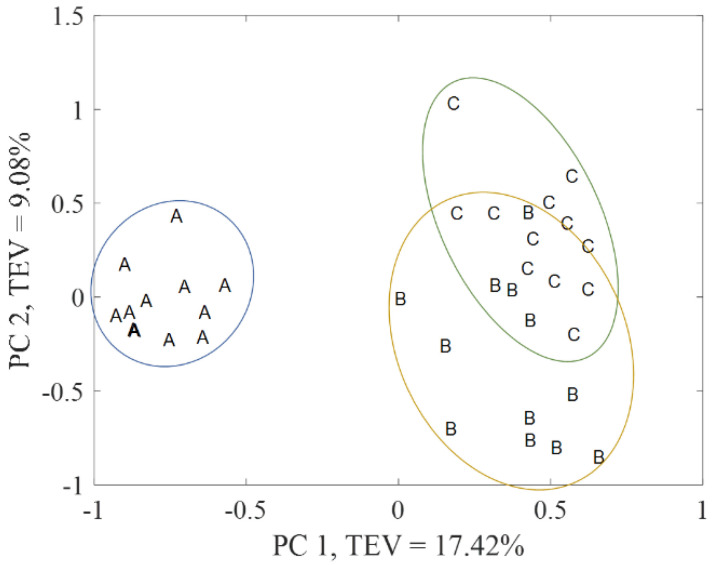
Super score plot of the MB-PCA model combining positive and negative ionization mode blocks. Piglet groups: newborns (A), infants (B), and children (C). TEV = total explained variance.

**Figure 4 metabolites-12-00739-f004:**
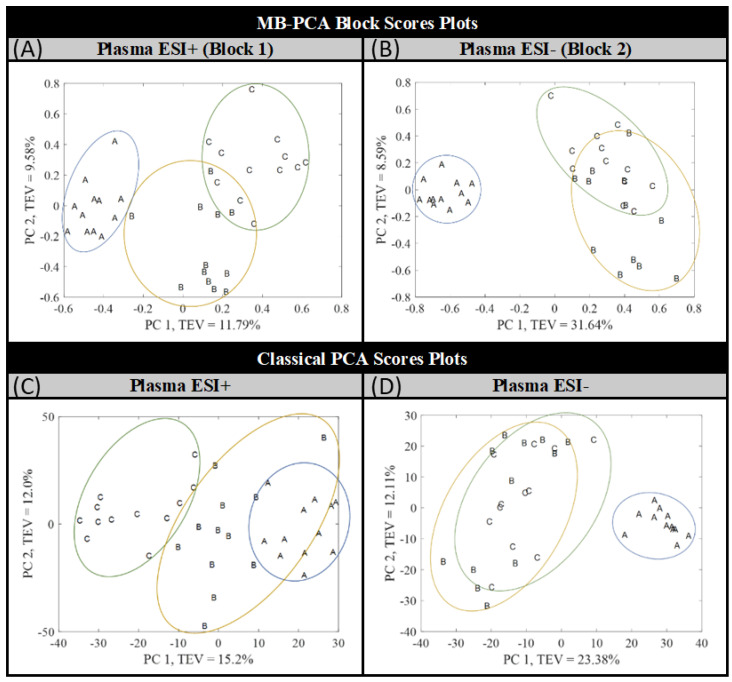
Upper score plots: Block score plot of the blocks (**A**) Plasma ESI+ and (**B**) Plasma ESI− of the MB-PCA model. Lower score plots: Score plot of classical PCA for the autoscaled matrix of (**C**) Plasma ESI+ and (**D**) Plasma ESI−. Piglet groups: newborns (A), infants (B), and children (C). TEV = total explained variance.

**Figure 5 metabolites-12-00739-f005:**
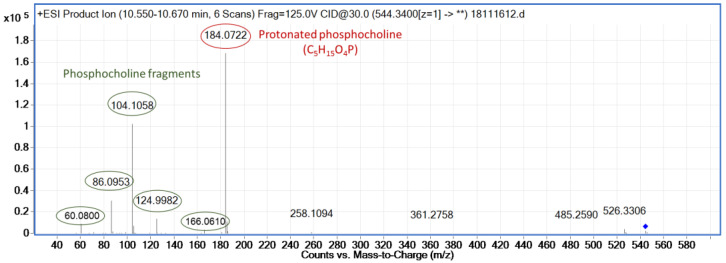
Experimental fragmentation pattern of the significant feature with *m*/*z* = 544.3400 and RT = 10.54 min, which has a phosphocholine in its structure.

**Figure 6 metabolites-12-00739-f006:**
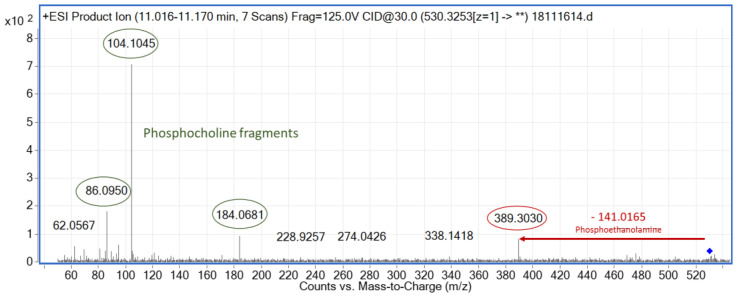
Experimental MS/MS spectrum for the significant feature in positive ionization mode with *m*/*z* 530.3253 and RT 11.04 min.

**Table 1 metabolites-12-00739-t001:** XCMS parameters for plasma samples at both ionization modes used in this study.

Algorithm	Parameter	ESI+	ESI−
CentWave	ppm	31.68	31
	peakwidth	22.01, 81.26	20, 80
	mzdiff	−0.0123	−0.0120
Obiwarp	profStep	0.7324	1
	gapInit	0.7552	0.9280
	gapExtend	2.400	2.688
Density	Bw	0.250	0.879
	mzwid	0.0270	0.0342

**Table 2 metabolites-12-00739-t002:** Number of features of interest in the Plasma ESI+ and Plasma ESI− data sets after applying univariate statistical analysis.

	Plasma ESI+	Plasma ESI−
Total number after matrix filtering	2207	1855
ANOVA and FDR (*p* < 0.001)	225	489
Post-hoc Tukey HSD test (A ≠ B ≠ C)	36	89
Fulfil normality	**26**	**73**
Do not fulfil normality	10	16
Kruskal–Wallis (*p* <0.001)	**1**	**1**
**Total significant features**	**27**	**74**

**Table 3 metabolites-12-00739-t003:** Significant features in plasma samples at positive and negative ionization modes that are considered real peaks after applying univariate analysis in the independent data sets. The table includes the *m*/*z* values, the retention time (RT), tendency, *q*-value (*p*-value after applying false discovery rate), the putative annotation, and the ion specie.

Plasma ESI+						Plasma ESI−					
*m*/*z*	RT (Min)	Regulation ^a^	*q*-Value	Annotation	Ion Specie	*m*/*z*	RT (Min)	Regulation ^a^	*q*-Value	Annotation	Ion Specie
400.1157	6.6	Up	1.2 × 10^−5^	Unknown		398.0972	6.5	Up	4.55 × 10^−9^	Unknown	-
364.0715	6.8	Up	4.43 × 10^−5^	Unknown		343.0242	6.8	Up	5.52 × 10^−8^	Unknown	-
271.9848	6.8	Up	6.24 × 10^−5^	Unknown		457.0161	6.8	Up	3.70 × 10^−9^	Unknown	-
212.5111	6.9	Up	1.06 × 10^−5^	Unknown		428.1105	7.7	Up	5.00 × 10^−6^	Unknown	-
211.0713	10.4	Up	5.86 × 10^−5^	Unknown		415.1959	8.0	Up	1.16 × 10^−9^	Unknown	-
200.2004	11.1	Up	2.35 × 10^−4^	Unknown		586.3141	10.0	Up	3.42 × 10^−7^	LPC (20:5)	[M-COOH]^−^
714.2590	9.3	Down	2.28 × 10^−5^	Unknown		615.3475	10.7	Up	5.34 × 10^−6^	LPC class	-
356.2795	9.7	Down	5.67 × 10^−5^	Acylcarnitine	-	411.2371	8.2	Down	1.00 × 10^−14^	Unknown	-
628.2926	10.5	Down	4.53 × 10^−5^	Unknown		350.2097	9.0	Down	2.68 × 10^−9^	Unknown	-
544.3400	10.5	Down	2.51 × 10^−4^	LPC (20:4)	[M+H]^+^	497.3464	9.4	Down	5.38 × 10^−8^	Unknown	-
300.6346	10.6	Down	1.74 × 10^−4^	Unknown		513.3004	9.9	Down	7.34 × 10^−5^	Unknown	-
530.3254	11.0	Down	4.34 × 10^−4^	LPE (22:4)	[M+H]^+^	447.3090	10.7	Down	1.51 × 10^−10^	Unknown	-
				LPC (17:1)	[M+Na]^+^	973.6249	10.9	Down	2.20 × 10^−6^	Unknown	-
548.3703	11.5	Down	1.58 × 10^−4^	LPC (20:2)	[M+H]^+^	478.2922	10.9	Down	1.72 × 10^−5^	LPE (18:1)	[M-H]^−^
						235.0707	5.6	Other	2.32 × 10^−4^	Unknown	-
						315.1055	6.0	Other	6.89 × 10^−6^	Unknown	-
						230.0111	6.5	Other	3.96 × 10^−6^	Unknown	-
						117.6456	8.1	Other	1.91 × 10^−8^	Unknown	-
						815.5669	8.9	Other	1.38 × 10^−4^	Unknown	-
						436.2815	11.0	Other	1.33 × 10^−7^	LPE (15:1)	[M-H]^−^
						526.3490	11.5	Other	5.49 × 10^−11^	Unknown	-

^a^ Up: Newborns (A) lower than infants (B) and children (C) (A < B < C). Down: Children (C) lower than infants (B) and newborns (A) (A > B > C). Other: Newborns (A) and children (C) lower than infants (B) (B > A and C).

## Data Availability

The MS data are available at the NIH Common Fund’s National Metabolomics Data Repository (NMDR) website, the Metabolomics Workbench, https://www.metabolomicsworkbench.org (accessed on 1 August 2022), where it has been assigned project ID PR000979. The data can be accessed directly via its project DOI: “http://dx.doi.org/10.21228/M84X3K”).

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
