# Peer review of "HPLC–(Q)-TOF-MS-Based Study of Plasma Metabolic Profile Differences Associated with Age in Pediatric Population Using an Animal Model"

_metabolites, 2022, doi:10.3390/metabo12080739_

Round 1

Reviewer 1 Report

Review of metabolites-1839550, by Albóniga et al.

The paper presents the differential assessment of plasma metabolomics from piglets of different ages: newborns (<5days), infants (2 week), sand children (4 weeks) with the final goal of understanding differences in drug administration and dosage in paediatrics.  The paper is well written, the experiment is well designed, and the analysis follows state-of-the-art procedures. I only have a few comments:

-          Experimental design: It would be useful to orthogonalize against the individual variability. To this extent, it would have been better to extract plasma from the same individuals at different time points (A, B, and C). Please, comment.

-          Outliers detection is most often performed using both the scores and residuals in PCA (using the D-statistic and Q-statistic). Please comment on how it was done and why.

-          The time evolution is an ordinal factor. It is not clear whether this is treated as ordinal or nominal (e.g., in PLS-DA). Please, discuss this issue and justify your decision.

-          I would drop the analysis on ESI- with log10. It would be enough to show autoscaling results, given the fact that models for ESI- for both autoscaling and log10 are presented, and that authors seem to suggest that autoscaling was more appropriate. Please clarify what preprocessing was used in PLS-DA and MB-PCA. Please, clarify in the paper why log10 was used in case you leave these results.

-          I would suggest using Procrustes rotation to present score plots, at least in the SM (some original plots should be maintained). This better allows to identify similarities/differences among models. A central choice here, which I could not find in the paper, is the procedure for dimensionality selection (choice in the number of LVs). Please, clarify.

-          A final suggestion with moderated interest: I would suggest using ASCA, as it allows a general modelling of your data. See for instance Díaz, C., et al. Predicting dynamic response to neoadjuvant chemotherapy in breast cancer: a novel metabolomics approach. Molecular Oncology, 2022. IMHO, ASCA allows to combine ANOVA and PCA in a much more powerful way than PLS-DA does. It is pretty simple to model cofounders (e.g., the individual) and ordinal variables is ASCA (the latter also in PLS-DA), and if the number of metabolites in ESI+ and ESI- is the same, you can also include this in the model as a confounder.

Reviewer 2 Report

In this work Albóniga et al. perform a metabolomics analysis of plasma samples of piglets as an animal model to explore the metabolic changes in newborn, infants and children. The work represents a novelty since metabolomics has been more focused on to study the onset, development and diagnosis of diseases, but not on the time-relateddevelopment of healthy individuals. They perform a correct analytical pipeline for metabolomics and in the analysis they try to improve the results incorporating multiblock PCA, which, although in their case it only produces a very subtle improvement of group separation, its interesting to introduce it in a metabolomics study, as a practical application in the case of different sets of omics data from the same sample. Also, regarding the identification of compounds, it is a shame that the authors have not be able to match their spectral data to identify the most part of the significant compounds. I encourage them to find alternatives, such as the open source software MSDIAL to process their MS/MS injections and find matches.

Some issues that have to be addressed and recommendations:

- Lines 102-109: Why is a mixture of drugs used as a system suitability test solution? Wouldn’t it be more suitable to use endogenous metabolites for this purpose?

- Line 124: Please use words, instead of PPT.

- Line 364-365. If at the end, 13 and 21 features are considered as real peaks, the numbers 27 and 74 for ESI+ and ESI-, respectively, should be changed in the abstract and in the conclusions to these referring to the real peaks.

- Table 5. Putative annotation of 615.3475 should be changed to LPC class (instead of lipid class, according to the text)

- To my understanding, Pearson Correlation Coefficient is used to help in the metabolite annotation in the case that two features have high correlation (>0.8) at the same retention time. In table S1 should be specified in which cases the two mass values of the two columns belong to the same molecule or not, and what are the potential adducts that relate both values. Maybe in some cases there are not the same molecule, but just molecules with strong relationship and the same behavior?

- Supplementary figure S4 it is not very informative, maybe it should be more focused on the variables that have strong correlation.

- Results of univariate analysis for the significant features should appear somewhere with the levels in each of the groups, at least as supplementary information.

- To reinforce the results about the changes observed in LPC and LPE, I suggest to investigate the levels of other LPC and LPE species with different chain lengths (the most common, from 16, to 24, with 0,1 and 2 insaturations) to see if the others follow the same pattern in each of the groups investigated, even if the changes in these species are not statistically significant.

- In Tables 4 and 5, for a better understanding of the results obtained with the significant features putatively identified, the up or down regulation between groups should also be shown, as it is done in table 3. According to this table, LPC (20:4) in ESI+ is down regulated, and LPC (20:5) in ESI- is up-regulated. This can be confusing, although I know that these are two different group comparisons. To clarify this and make it more understandable I suggest a graphical representation such as a heatmap.  Maybe other non-significant species that I mentioned in the previous point, could be included on it.

Reviewer 3 Report

I thought this research was clearly presented and contained some robust statistics.

Just a couple of suggestions. Tables 3, 4 and 5 should be combined to include the putative annotations. A features table on it's own isn't really adding anything to add to the story.

The discussion/conclusion needs more information as to how the lipid distribution affects organ maturation and how this study has helped to answer your question. Although a good summary of the findings it would be improved with some discussion about the mechanisms leading to changes in lipid distribution/concentration and how these link with maturation.

Round 2

Reviewer 2 Report

The main issues have been adressed by the authors. Now is suitable for publication